# Polarizing Macrophage Functional Phenotype to Foster Cardiac Regeneration

**DOI:** 10.3390/ijms241310747

**Published:** 2023-06-28

**Authors:** Claudia Molinaro, Mariangela Scalise, Isabella Leo, Luca Salerno, Jolanda Sabatino, Nadia Salerno, Salvatore De Rosa, Daniele Torella, Eleonora Cianflone, Fabiola Marino

**Affiliations:** 1Department of Medical and Surgical Sciences, Magna Graecia University, 88100 Catanzaro, Italy; claudia.molinaro@studenti.unicz.it (C.M.); saderosa@unicz.it (S.D.R.); 2Department of Experimental and Clinical Medicine, Magna Graecia University, 88100 Catanzaro, Italy; m.scalise@unicz.it (M.S.); isabella.leo@unicz.it (I.L.); l.salerno@unicz.it (L.S.); sabatino@unicz.it (J.S.); nadia.salerno@unicz.it (N.S.); marino@unicz.it (F.M.)

**Keywords:** macrophages, innate immunity, cardiac regeneration, inflammation

## Abstract

There is an increasing interest in understanding the connection between the immune and cardiovascular systems, which are highly integrated and communicate through finely regulated cross-talking mechanisms. Recent evidence has demonstrated that the immune system does indeed have a key role in the response to cardiac injury and in cardiac regeneration. Among the immune cells, macrophages appear to have a prominent role in this context, with different subtypes described so far that each have a specific influence on cardiac remodeling and repair. Similarly, there are significant differences in how the innate and adaptive immune systems affect the response to cardiac damage. Understanding all these mechanisms may have relevant clinical implications. Several studies have already demonstrated that stem cell-based therapies support myocardial repair. However, the exact role that cardiac macrophages and their modulation may have in this setting is still unclear. The current need to decipher the dual role of immunity in boosting both heart injury and repair is due, at least for a significant part, to unresolved questions related to the complexity of cardiac macrophage phenotypes. The aim of this review is to provide an overview on the role of the immune system, and of macrophages in particular, in the response to cardiac injury and to outline, through the modulation of the immune response, potential novel therapeutic strategies for cardiac regeneration.

## 1. Introduction

Cardiovascular diseases (CVDs) represent the leading cause of death worldwide, accounting for about 31% of all deaths [1]. The recent technological and therapeutical advances, along with a better understanding of the pathophysiological mechanisms involved in several CVDs, have led to the increased survival rates of these patients. This is however counterbalanced by an increased number of patients suffering from the sequalae of an acute cardiovascular event. For instance, patients with a previous acute coronary syndrome are still at risk of developing heart failure (HF) due to the presence of reparative fibrosis and consequent adverse cardiac remodeling with impaired function [2]. The adult heart harbors multiple heterogeneous cellular components, including cardiomyocytes (CMs), fibroblasts, smooth muscle cells, endothelial cells, cardiac stem cells, pericytes, and a plethora of immune cells [3], as shown in Figure 1. The latter are actively involved in the inflammatory response that follows cardiac injury in attempts to antagonize and repair myocardial damage and restore cardiac homeostasis. The goal is the clearance of the fibrotic tissue with the initiation of a reparative cascade preventing adverse myocardial tissue remodeling. Among all the innate immune cells, macrophages are specifically involved in the onset and resolution of inflammation. Their dysregulation is a primary contributor to tissue inflammaging, a pro-inflammatory status associated with high levels of pro-inflammatory markers. Tissue-resident C-C chemokine receptor 2 (CCR2^−^) and tissue-resident/systemically recruited CCR2^+^ cardiac macrophages differentially affect cardiac remodeling and repair following myocardial injury [4] (Figure 1). An exhaustive comprehension of the different responses induced by the two subsets of cells is key for the development of new therapeutic strategies to prevent fibrosis and adverse remodeling and promote the formation of new functional myocardium. The intrinsic regenerative potential of the adult heart after an injury is in fact significantly limited, at least in response to ischemic damage [5,6]. It has been demonstrated that this endogenous potential can be fostered after an ischemic injury using several approaches, from the administration of exogenous cell therapy to RNA therapeutics [7,8,9]. However, this reparative response may be further reduced by the concomitant presence of cardiovascular risk factors (i.e., aging or diabetes) that create an adverse cellular microenvironment halting regeneration [7,10,11,12,13,14,15,16]. In this scenario, cell therapy was intended as an alternative strategy to restore/replace the damaged and dysfunctional cardiac tissue to improve cardiac function [7,17,18]. The immune response and the endogenous cardiac repair system interact to modulate damage resulting from inflammatory response. It is still an open question however as to whether specific aspects of the immune/inflammatory response are responsible for a predominantly fibrotic and poorly regenerative response to injury in the adult heart and/or whether their modulation could positively affect the regenerative response. The aim of this review is to provide a comprehensive analysis about the interaction between the immune response and the cardiac reparative/regenerative process, focusing on the specific role of the cells involved in this tangle.

## 2. Innate Immunomodulation after Injury

Insults that induce cardiomyocyte death promote the activation of the immune response to restore tissue integrity [19]. The immune system is involved in damage-associated signaling, inflammation, revascularization, and fibrotic scar formation [19]. Macrophages play a key role in all the stages of the immune response, with distinct phenotypes performing specific functions at different time points [20]. It is known that macrophages derived from monocytes can be further classified into two main types: the pro-inflammatory M1 type and the resolving M2 type [21]. However, it is also widely accepted that the M1/M2 paradigm is just a simplification, and the exact sources and phenotypes of macrophages are yet to be fully clarified. In addition, it still remains unclear if M1 macrophages can switch to an M2 type, representing a mixed phenotype, or if these two subsets of macrophages necessarily originate from completely different sources [22]. Nevertheless, it has been established that the cellular response to heart damage can be divided into three distinct phases: the inflammatory, proliferative, and resolutive phases (Figure 2) [23].

### 2.1. Inflammatory Phase

Cardiomyocyte death and the subsequent release of intracellular components into the extracellular compartment trigger the inflammatory response, attracting resident immune cells. These components, known also as Damage-Associated Molecular Patterns (DAMPs), include nucleic acid fragments, heat shock proteins, adenosine triphosphate (ATP), and fragmented extracellular matrix (ECM) components. They primarily activate the innate immune pathways and the inflammatory response through Toll-like receptors (TLRs) and NOD-like receptors (NLRs), which are expressed on both cardiomyocytes (CMs) and resident immune cells [24,25]. This leads to the release of specific pro-inflammatory cytokines, particularly Interleukin (IL)-1β and IL-18 [26].

Furthermore, DAMPs activate the complement system, which recognizes and subsequently destroys damaged cells through phagocytosis [27]. Uncontrolled activation and amplification of the complement cascade can be detrimental, resulting in significant tissue damage. However, the neutralization of complement can reduce myocardial injury and mortality in patients with myocardial infarction (MI) [28].

After an injury, vascular endothelial cells increase the expression of endothelial intercellular adhesion molecule 1 (ICAM-1) and vascular cell adhesion molecule 1 (VCAM-1), while histamine released by mast cells increases vascular permeability. Both of these phenomena contribute to facilitating leucocyte infiltration. Consequently, multiple cellular effectors such as neutrophils and monocytes are attracted to the sites of damage. Neutrophils are the first to migrate, followed by monocytes, which subsequently differentiate into M1-type macrophages. The reactive oxygen species (ROS) generated by activated neutrophils [29] promote the infiltration and proliferation of monocytes, dendritic cells, natural killer (NK) cells, T helper cell type 1 (Th1), T helper cell type 17 (Th17), B-lymphocytes, and additional neutrophils [30]. Additionally, neutrophils serve as a source of matrix-degrading enzymes and are responsible for the phagocytosis of the degraded matrix components and cells coated in complement opsonin [31].

Neutrophil activity can also contribute to exacerbating injury. The secretion of myeloperoxidase (MPO) by neutrophils, in fact, leads to maladaptive cardiac remodeling after injury. For instance, deletion of MPO in adult wild-type (WT) mice results in decreased left ventricular (LV) dilatation and a significant improvement in LV function compared to the control group [32]. Furthermore, the depletion of endothelial Brahma-related gene 1 (Brg1), which mediates neutrophil–endothelium adhesion, results in decreased ventricular fibrosis, reduced infarct size, and better recovery of cardiac function [33]. Inhibition of certain neutrophil-derived enzymes and the reduction of neutrophil infiltration achieved through antibody-mediated blockage of specific adhesion molecules have been shown to decrease tissue damage following MI and reperfusion [34]. Monocytes represent the second type of immune cells involved in the inflammatory phase. Interferon (IFN)-γ, Tumor Necrosis Factor (TNF)-α, and DAMPs promote monocyte recruitment and their differentiation into M1-type macrophages, which are inflammatory macrophages. These M1 macrophages, in turn, release other pro-inflammatory factors such as TNF-α, IL-1β, chemokine C-X-C motif ligand 10 (CXCL10), IL-6, IL-12, and IL-23 [35,36]. As with neutrophils, monocytes also release matrix metallopeptidase 9 (MMP-9) that degrades basement membranes, thereby facilitating the recruitment of additional immune cells [37].

### 2.2. Proliferative Phase

The inflammatory phase ends when M1 macrophages phagocytose neutrophils [38]. At this stage, M1 macrophages decrease their production of pro-inflammatory factors and begin to increase the secretion of two anti-inflammatory factors, namely IL-10 and transforming growth factor beta (TGF-β). This represents a signal of a shift towards the M2 phenotype. It is important to note that macrophages likely exist along a spectrum of mixed phenotypes [39] and exhibit plasticity [40]; thus, the M1/M2 paradigm may be an oversimplified and inaccurate classification. Analysis using single-cell RNA sequencing (sc-RNA seq) has revealed the presence of distinct macrophage phenotypes associated with regenerative and fibrotic processes [41]. Regardless, macrophages operating at this stage promote angiogenesis by secreting IL-10, TGF-β, and vascular endothelial growth factor (VEGF) [42], as well as activate fibroblasts [43]. TGF-β stimulates newly formed myofibroblasts to secrete collagen (predominantly type III), fibronectin, and other extracellular components [44]. Additionally, macrophages regulate matrix turnover through the modulation of MMPs and tissue inhibitors of metallopeptidases (TIMPs) [45]. As a result of all these processes, a temporary collagenous matrix enriched in fibrin and fibronectin is formed [44,45].

### 2.3. Resolutive Phase

The final phase is the resolutive phase, during which a remodeling process occurs involving the replacement of newly laid type III collagen with type I collagen. Type I collagen becomes cross-linked, resulting in a stronger scar with increased tensile strength. The matrix becomes less populated with cells as most of the remaining leukocytes undergo apoptosis. However, some myofibroblasts may persist, and their presence can have a negative impact on cardiac function due to their distinct electrical properties compared to CMs [46].

The exact mechanisms that trigger this phase are not yet fully understood. Typically, the described events progress through a series of cascade steps involving the reduction of pro-inflammatory factors, cessation of granulocyte recruitment, and an increase in monocyte levels. Monocyte-derived macrophages play a crucial role in removing inflammatory cells and tissue debris, leading to the resolution of inflammation and promotion of the recovery of cardiac structure and contractile function [47,48].

## 3. Adaptive Immune Response

In addition to the innate immune response, the release of DAMPs triggers an adaptive immune response, which involves the activation of B-lymphocytes and T-lymphocytes. Moreover, in the presence of an inflammatory environment, cardiac self-antigens such as myosin and troponin can disrupt the tolerance mechanism, leading to the activation of long-lived antigen-specific adaptive immune cells [49,50]. While the innate immune response is regulated by negative feedback mechanisms aimed at resolving early inflammation, the adaptive immune response does not appear to be controlled in the same manner. The release of substantial amounts of self-antigens can ultimately result in autoimmune tissue damage, leading to the subsequent release of more self-antigens.

The precise role of B-lymphocytes in this phase is still not fully understood, although there is evidence suggesting their involvement in the immune response to cardiac injury. Mice with depleted B-lymphocytes have shown improved cardiac function after MI [51]. Furthermore, activated B-lymphocytes produce pro-inflammatory cytokines such as TNF-α which directly contribute to myocardial dysfunction by reducing contractility, inducing myocyte apoptosis [52], and promoting fibroblast differentiation into myofibroblasts [53]. Similarly, the exact function of T-lymphocytes in response to cardiac injury has not been completely elucidated. The rapid changes in the local environment following an insult lead to the emergence of a heterogeneous subpopulation of T-lymphocytes which, depending on the timing and type of cardiac damage, have varying effects that can either have a positive or detrimental impact on the healing process [54]. Specifically, regulatory T-lymphocytes appear to play a beneficial role in the immediate post-MI phase [55,56]. Conversely, it has been demonstrated that CD4^+^ T cells, but not CD8^+^ T cells, contribute to myocardial ischemia–reperfusion injury through the release of IFN-γ [54].

Indeed, in the context of cardiac injury, the adaptive immune response can have an overall negative effect, potentially leading to further tissue damage. The continuous exposure to self-antigens can trigger persistent immune autoreactivity that may eventually involve previously unaffected cardiac regions of the heart. This process can contribute to negative remodeling and LV dilation.

## 4. Distinct Cardiac Macrophage Subsets among the Adult Heart

The adult mammalian heart harbors heterogeneous populations of macrophages, each originating from distinct developmental pathways [57,58]. Macrophages serve as resident immune cells in the tissue and play a crucial role in maintaining tissue homeostasis [59]. Following an injury, the recruitment of specific macrophage subsets influences the cellular microenvironment and the resulting response to cardiac injury. These macrophage subsets can either contribute to maladaptive remodeling through a pro-inflammatory response [60] or promote reparative processes, including cardiac regeneration [21,61]. As mentioned earlier, macrophages are typically classified into two types: M1 and M2. M1-type macrophages are classically activated by factors such as IFN-γ and lipopolysaccharide (LPS), leading to a pro-inflammatory response [62]. On the other hand, M2-type macrophages are alternatively activated in response to cytokines such as IL-4 and IL-13, exhibiting a resolving phenotype in vitro [63].

However, recent experiments utilizing genetic lineage tracing and fate mapping have challenged the simplistic classification of macrophages into M1 and M2 types. These findings suggest that the polarization process of macrophages may be more complex and dynamic than previously thought, and the observed markers on in vitro-generated macrophages may not accurately reflect the phenotype of macrophages in vivo, particularly in classically activated mice models [64]. Furthermore, it is important to differentiate between macrophages derived from circulating monocytes and tissue-resident macrophages, which exhibit tissue-specific features [57,58]. Tissue-resident macrophages originate from the yolk sac or fetal monocyte progenitors [65], and they are ontogenetically older than macrophages derived from the bone marrow [66]. These tissue-resident macrophages are evolutionarily conserved throughout the lifespan [67]. In the murine heart, resident cardiac macrophages constitute around 5–10% of the non-myocyte population, and their percentage increases following cardiac damage [68]. Within cardiac tissue, at least two distinct macrophage subsets and one monocyte subset have been identified based on the presence or absence of the CCR2 receptor on their surface. CCR2 negative (CCR2^−^) cells originate from yolk sac progenitors and are detected in the cardiac tissue around embryonic day 12.5 (E12.5) [58]. These cells are primarily located within the myocardial wall and in close proximity to the coronary vasculature. On the other hand, CCR2 positive (CCR2^+^) cells are derived from fetal monocyte progenitors and can be found in the trabecular projection of the endocardium starting from E14.5 [58]. CCR2^−^ cells, also known as a “resident macrophage population”, are self-regenerating macrophages that do not require prior monocyte recruitment. CCR2^+^ cells, derived from hematopoiesis, are maintained through monocyte recruitment and are referred to as the “non-resident macrophage” population derived from circulating monocytes [69]. Although cardiac macrophages can exhibit M1 or M2 phenotypes in response to various stimuli, it is important to note that this phenotypic expression may not be permanent. Studies have shown that cardiac macrophages, primarily exhibiting an M2-like phenotype, can transition to an M1-like phenotype in aged mice [70]. Moreover, tissue-specific gene expression in cardiac macrophages can be significantly altered in response to injuries such as MI, stroke, or sepsis [70].

In mice, the expression of CCR2, major histocompatibility complex (MHC) class II, and CD11c allows for further differentiation into three subtypes of cardiac macrophages [57]. There are two predominant CCR2^−^ populations: MHCII^high^/CD11c^low^ and MHCII^low^/CD11c^low^. These subsets are derived from yolk sac progenitors and are renewed through in situ proliferation. Additionally, there is a CCR2^+^, MHCII^high^, and CD11c^high^ subset, which is slowly replaced by circulating monocytes. Furthermore, there is one monocyte subset characterized by CCR2^+^/MHC-II^low^ expression. Transcriptomic analysis of these subsets reveals both overlapping and non-overlapping functions. CCR2^+^ macrophages exhibit a significant number of genes involved in the inflammatory process, suggesting pro-inflammatory activity [57,58,71]. The MHCII^high^ subsets have genes involved in antigen presentation to T cells, indicating a potential role in immune surveillance [57,58,71]. The CCR2^−^/MHCII^low^ subset has demonstrated uptake of apoptotic/necrotic cells, indicating a role in the clearance of dead cells and prevention of immune response [71]. It is important to note that these subsets represent only a part of the cardiac macrophages described in the literature. Additional studies have identified four distinct cardiac macrophage clusters with unique functions at steady state [72]. For example, CCR2^−^/TIMD4^+^/LYVE1^+^/MHCII^low^ corresponds to the CCR2^−^/MHCII^low^ subset and expresses genes involved in homeostasis and regeneration [58]. Similar CCR2 macrophage subsets have also been identified in the human heart, suggesting comparable functions [73]. In the human heart, distinct subsets of monocytes and macrophages can be identified based on the expression of CCR2 and HLA-DR [73]. Human cardiac macrophages are characterized by the co-expression of CD14, CD45, and CD64 markers. Within the CD14^+^/CD45^+^/CD64^+^ population, three subsets can be distinguished based on the expression of the human homologue of MHC-II (HLA-DR) and CCR2: CCR2^+^HLA-DR^low^, CCR2^+^HLA-DR^high^, and CCR2^−^HLA-DR^high^ cells [73]. It is important to note that there are differences between mouse and human macrophages. In mice, CCR2^−^ macrophages are divided into MHCII^low^ and MHCII^high^ subsets, while in humans they are predominantly HLA-DR^high^.

## 5. Cardiac Macrophage Recruitment following Myocardial Injury

In the response to cardiac injury, both cardiac and non-cardiac macrophages play a role. Following acute MI in mice, there is a significant reduction of approximately 60% in the number of resident CCR2^−^ macrophages in the infarcted area within 2 days of the event [74]. These resident macrophages are replaced by inflammatory CCR2^+^ monocytes and monocyte-derived macrophages. The role of these monocyte-derived macrophages is to promote monocyte recruitment through the production of C-C motif ligand 2 (CCL2), overcoming the inhibitory effect exerted by resident CCR2^−^ macrophages. While these monocyte-derived macrophages have a pro-inflammatory function, their production of inflammatory factors is generally lower than that of recruited CCR2^+^ macrophages [4]. To differentiate cardiac CCR2^+^ macrophages from circulating monocytes and monocytes-derived macrophages, the expression of type I IFN-stimulated genes can be used. This suggests that CCR2^+^ macrophages are responsive to type I IFN produced during myocardial infarction.

In murine models, circulating monocytes consist of two subsets: classical pro-inflammatory monocytes expressing high levels of lymphocyte antigen 6 complex (Ly6C) (or CD14^high^/CD16^−^ in humans) that are recruited to sites of inflammation and non-classical Ly6C^low^ monocytes (or CD14^low^/CD16^+^ in humans) that survey the luminal surface of vascular endothelial cells [75,76,77].

Indeed, both Ly6C^high^ and Ly6C^low^ monocytes participate in the immune response observed in the infarcted heart, representing distinct phases of monocytes recruitment [75,78]. During the initial inflammatory phase, there is a peak in the recruitment of Ly6C^high^ monocytes observed at around 3 days post infarction, followed by a gradual decline. These Ly6C^high^ monocytes express CCR2 and migrate into the injured site in response to the chemokine CCL2. Upon migration, they differentiate into recruited CCR2^+^ macrophages. CCL2 plays a crucial role in the recruitment of the Ly6C^high^ monocyte into the infarcted area, and CCL2-deficient (CCL2^−^/^−^) mice have shown reduced monocyte infiltration, interstitial fibrosis, and ventricular dysfunction in response to myocardial ischemia compared to WT mice [79]. Once recruited, CCR2^+^ macrophages contribute to the immune response by releasing pro-inflammatory factors and matrix metalloproteinases that facilitate the degradation of the extracellular matrix and removal of debris and necrotic cells. Therefore, a reduction in circulating monocytes can result in the accumulation of uncleared debris, necrotic tissue, and myocardial fibrosis [75].

Ly6C^low^ monocytes appear later in the response to cardiac injury, with a peak observed at around day 7 post injury, and they represent a significant portion (approximately 75%) of the total macrophage population by day 16. These Ly6C^low^ monocytes give rise to reparative and non-inflammatory macrophages. Ly6C^low^ monocytes express high levels of growth factors such as VEGF [75], which are involved in promoting myofibroblast accumulation, angiogenesis, and collagen deposition [78]. Previous studies have proven that two different pathways are responsible for the accumulation of Ly6C^low^ macrophages in the damaged area [75]. One pathway is related to the expression of C-X3-C Motif Chemokine Receptor 1 (CX3CR1) in the injured zone. Additionally, it is possible that Ly6C^high^ monocytes directly differentiate into proliferating Ly6C^low^ macrophages within the myocardium, driven by the induction of the orphan nuclear receptor Nr4a1 [78]. Depletion of Nr4a1 results in increased expression levels of CCR2 among cardiac Ly6C^high^ monocytes, leading to the induction of macrophages with high pro-inflammatory activity. The lack of Nr4a1 has also been associated with impaired LV function after myocardial infarction, limited cardiac healing, enhanced myocardial scar size, and reduced collagen density [78]. Therefore, Nr4a1 plays a crucial role in regulating the differentiation and function of Ly6C^low^ macrophages in the context of cardiac injury.

## 6. Macrophages and Cardiac Tissue Regeneration

The debate regarding cardiac regeneration and the existence and role of endogenous resident cardiac stem cells (CSCs) is indeed ongoing. While the regenerative capacity of skeletal muscle mediated by satellite cells is well established, the presence and phenotypic characterization of CSCs in the heart are still subjects of investigation [15]. In skeletal muscle regeneration, the process begins with an inflammatory response triggered by an insult. This leads to the activation, differentiation, and fusion of satellite cells, including muscle stem cells. These activated satellite cells contribute to the growth and remodeling of newly formed myofibers. Interestingly, a small portion of myogenic precursor cells does not undergo terminal differentiation, instead remaining as a pool of stem cells that can be utilized for future regeneration if needed [80]. It has been demonstrated that the complete elimination of the satellite cell pool, specifically all Pax7^+^ cells, in adult skeletal muscle suppresses muscle regeneration entirely [81].

Indeed, macrophages have been implicated in the fate of skeletal muscle satellite stem cells during regeneration [82]. While the exact role of resident macrophages in this context is not yet fully understood, studies have shown their involvement in regulating tissue homeostasis under normal conditions. However, they appear to have limited phagocytic capacity during injury and instead act as sentinels, becoming activated in response to DAMPs and promoting the recruitment of circulating leukocytes [83]. Similar to the observations in cardiac injury, the recruitment of monocytes/macrophages in skeletal muscle regeneration follows a sequential pattern [84,85]. The pro-inflammatory Ly6C^high^ monocytes are the first population recruited during the acute phase of inflammation, while the anti-inflammatory Ly6C^low^ subset appears later [86]. It has been demonstrated that Ly6C^low^ macrophages can arise from the Ly6C^high^ subset [84,85], as evidenced by an in vivo experiment showing a nearly complete transition to the Ly6C^low^ phenotype by day 3 after acute injury [85]. The sequential presence of Ly6C^high^ and Ly6C^low^ macrophages is associated with specific events in the regenerative process [87]. One study demonstrated numerous regenerating areas seven days after an injury, characterized by the presence of proliferating cells (ki67^+^/CD56^+^) and/or differentiating cells (myogenin^+^) [82]. These areas also exhibited a positive presence of both pro-inflammatory macrophages (identified by the expression of iNOS and COX2 in CD68^pos^ cells) and anti-inflammatory macrophages (marked with CD206 and CD163). Specifically, pro-inflammatory markers were commonly expressed by macrophages in regenerating areas containing only myogenin^−^ cells compared to those containing at least one myogenin^+^. Conversely, Arg1 macrophages (another anti-inflammatory marker) were more abundant in regenerating areas containing myogenin^+^-proliferating myogenic precursor cells (MPCs) compared to those lacking differentiating MPCs. Collectively, these findings suggest a preferential association between proliferating MPCs and macrophages expressing pro-inflammatory markers, while regenerating areas with differentiating myogenin^+^ MPCs tend to be associated with anti-inflammatory macrophages.

There is growing interest in research focused on promoting cardiac regeneration through stem cell transplantation or the induction of endogenous CSCs [88,89]. While pre-clinical and clinical trials have reported beneficial effects of stem cell therapy on infarcted hearts, there are currently limited data regarding stem cell survival after transplantation and their ability to generate new functional myocytes [90]. It has been suggested that the pro-inflammatory environment following cardiac injury can lead to damage to transplanted stem cells, primarily through the stimulation of apoptosis, necrosis, and autophagy cascades [91,92]. Additionally, the pro-inflammatory cytokines present in this environment may contribute to the failure of stem cells to commit to the cardiac lineage [93]. However, it appears that the functional benefits observed with injected cell therapy are related to an acute inflammation-based wound-healing response [94]. Vagnozzi et al. conducted experiments on healthy mice to assess the effects of two types of primary adult cells, both of which contained small fractions of true stem cells: fractionated bone marrow mononuclear cells (MNCs) and cardiac mesenchymal cells. These cells were administered either as living cells or as dead cells, previously killed through a freeze–thaw cycle. In their findings, Vagnozzi et al. observed a temporary and regional induction of CCR2^+^ and CX3CR1^+^ macrophages following the injection of cells, regardless of cell type or viability. This led to a shift in the composition of macrophage subsets, transitioning from a predominant population of CX3CR1^+^/CCR2^−^ macrophages in the naive state to a mix of CCR2^+^ and CCR2^+^/CX3CR1^+^ macrophages. The experiment was also conducted on mice with induced myocardial infarction one week after ischemia–reperfusion (I-R) injury. Injection of mononuclear cells, cardiac mesenchymal cells, or zymosan (a yeast-derived protein–carbohydrate complex capable of inducing sterile inflammation) into the infarct border zone improved LV contractility and decreased end-systolic volume. These beneficial effects were associated with the selective activation of innate immune responses. Importantly, the positive effects of cell injection on infarcted mice were abolished when a broad spectrum of immunosuppressant agents or macrophage depletion was co-administered. However, the experimental design employed by Vagnozzi et al. did not allow for the detection of new cardiomyocyte formation [94]. However, other studies have demonstrated the generation of new cardiomyocytes following injury [23,95]. For instance, an acute cardiomyocyte loss caused by an isoproterenol overdose (ISO) activates the resident cardiac c-kit^+^ stem/progenitor cells (CSCs), leading to the generation of new cardiomyocytes that replace those lost after ISO-induced damage [23].

Similar to what is observed in skeletal muscle, macrophages are also known to be involved in cardiac regeneration in neonatal mice, which exhibit regenerative potential and can fully regenerate following MI [19,61,96]. Cardiac macrophage subsets in mice differ between adults and neonatal mice. Neonatal hearts host only one embryonal-derived macrophage subset of CCR2^−^/MHC-II^low^ and one monocyte CCR2^+^ population. In response to injury, neonatal mice selectively expand CCR2^−^/MHC-II^low^ macrophages without recruiting additional CCR2^+^ monocytes. These CCR2^−^ macrophages isolated from injured neonatal hearts produce lower levels of pro-inflammatory mediators. In contrast, in adult mice hearts, the CCR2^−^/MHC-II^low^ population is rapidly lost and replaced by pro-inflammatory monocytes and monocyte-derived macrophages expressing CCR2^+^/MHC-II^high^, characterized by limited capacity to promote cardiac repair and to generate inflammation or oxidative stress [21] (Figure 3). Assessing changes in epigenetic regulation through cardiomyocyte development has been of interest because of the drastic change in cardiomyocyte proliferation ability after the first few days of life [97]. Changes in epigenetics have proven to vary from neonatal proliferative-competent to adult terminally differentiated cardiomyocytes [97]. Furthermore, loss of heart regenerative capacity in adult versus neonatal mammals is triggered by increasing thyroid hormones and may be a trade-off for the acquisition of endothermy [98].

Finally, with aging, myocardial T cells undergo clonal expansion and exhibit an upregulated pro-inflammatory transcription signature, marked by increased IFN-γ production [99]. Physiological T-cell development or adoptive transfer of adult IFN-γ-producing T cells into neonatal infarcted mice shifted them toward an adult-like healing phenotype with monocyte-derived macrophage recruitment, contributing to impaired cardiac regeneration and promoting irreversible structural and functional cardiac damage [100]. These findings suggest a trade-off between myocardial regenerative potential and the development of T-cell competence and allow us to postulate that immunosenescence may account for the deficit in regenerative capability with age.

Aurora et al. conducted a study comparing the immune response of mice at different regenerative time points following MI and identified differences in the power and kinetics of the monocyte and macrophage response to injury [61]. Macrophages can secrete a plethora of soluble factors that may contribute to the formation of new cardiomyocytes [61]. Their depletion instead promotes fibrotic scar formation, resulting in reduced cardiac function and angiogenesis in neonatal mice.

Other experimental in vitro observations based on cell tracking strategies or tissue-specific gene depletion demonstrated the involvement of macrophages in skeletal muscle regeneration in both mice and human models [82]. It has been postulated that the recruitment of Ly6C^+^ monocytes/macrophages stimulates the quiescence niche of muscle stem cells (MuSCs), promoting their proliferation and preventing premature fusion of myogenic cells. It seems that during the resolutive phase, macrophages reduced inflammation levels while boosting stem cell angiogenesis, differentiation, and matrix remodeling [101].

Further studies are needed to determine whether the beneficial effects of cardiac macrophages on cardiac remodeling can contribute to enhancing the regenerative potential of CSCs. It is also important to investigate the potential link between macrophage activation/polarization and the fate of stem cells.

## 7. Macrophages and Tissue Degeneration during Aging

Aging is a complex phenomenon that involves several physiological changes as well as the immune system [102,103].

Cell senescence, although often used interchangeably with aging, is instead characterized by multiple hallmarks, including progressive accumulation of DNA damage, mitochondrial dysfunction, apoptosis, telomere shortening, oncogene activation or inactivation, epigenetic alterations, and ROS accumulation [104,105]. Senescent cells acquire a senescence-associated secretory phenotype (SASP) that involves the secretion of a wide range of soluble molecules, varying on the basis of the cell type and the triggering factor [10,16].

The combination of changes affecting the immune system during aging is known as immunosenescence and is characterized by the presence of a low-grade inflammation (inflammaging) that modulates macrophage activity and phenotype expression [106,107]. Interestingly, inflammaging is a common feature of different age-related diseases such as cardiovascular disease [16,108], type 2 diabetes mellitus, and diabetic cardiomyopathy [11,13,109], conditions all characterized by a loss of cardiac regenerative potential [10,11,13,14,16].

Aging can also negatively affect the ability to mitigate inflammation following a cardiac injury in murine models through the deregulation of certain metabolic pathways [110]. In aged mice, the clearance of senescent cells has been found to improve cardiac remodeling and function after myocardial infarction [7], which was also observed in animal models of diabetic cardiomyopathy [13]. Furthermore, aged hearts dramatically change the landscape of their leukocyte population with more monocyte-derived cardiac macrophages, though they are smaller in size and with larger granulocytes [111].

In addition, the SASP factors can shift macrophage polarization from an anti-inflammatory phenotype to a pro-inflammatory one [112]. This is in line with the linear increase in cardiac macrophages with a pro-inflammatory phenotype observed with aging [36], that may be a result of uncontrolled monocyte recruitment, alterations in monocyte fate determination, or changes in resident macrophage behavior [36].

Resident cardiac macrophages in fact exhibit a reduction in self-renewal ability that is maintained, at least in part, by the increased contribution of macrophages derived from circulating monocytes [111]. Similarly, macrophages that are implicated in skeletal muscle regeneration underwent significant changes during aging [113]. Their release of proliferative factors is in fact impaired, with consequences on satellite cell function and muscle regeneration [113]. Previously, Wang et al. demonstrated that the supernatant obtained from old bone marrow-derived macrophages (BMDMs), compared to that obtained from young BMDMs, has a reduced number of proliferative Ki67^+^ myoblasts [114]. However, the observed effects of aging on macrophages are widely influenced by the marker and the experimental model used, as well as by the subtype of population examined. For example, an increase in anti-inflammatory macrophages (marked as CD68^+^/CD163^+^) has been described in resting muscles of aged mice, correlating with an increase in skeletal muscle fibrosis [115,116]. Another study instead demonstrated a decrease in the number of both pro-inflammatory (CD11b^+^) and anti-inflammatory macrophages (CD163^+^) in old subjects compared to young controls (average 71.4 years vs. 31.9 years) [117].

## 8. Therapeutic Perspectives

All the evidence discussed so far seems to open new interesting therapeutical approaches based on the modulation of macrophage function. Several data indicate that a switch towards the cardioprotective anti-inflammatory phenotype can improve cardiac repair and function after injury [118,119,120].

In a rat model of acute MI, treatment with phosphatidylserine (PS)-presenting liposomes (mimicking the anti-inflammatory effects of apoptotic cells) induced increased release of anti-inflammatory cytokines such as TGFβ and IL-10, along with concomitant downregulation of the pro-inflammatory markers TNFα and CD86, in macrophages in both in vivo and in vitro models. This treatment supported angiogenesis and prevented ventricular dilatation and remodeling [118].

Additionally, some stem cell therapies also appear to have positive effects on recovery after myocardial damage through macrophage modulation. For example, bone marrow-derived mesenchymal stem cells (BM-MSCs) are able to modify their macrophage phenotype toward an M2-like status. Infarcted mice treated with BM-MSCs exhibited increased cardiac expression of F4/80 + CD206 + macrophages and demonstrated an improvement in cardiac function, as well as a reduction in pro-inflammatory factors and an increase in the expression of anti-inflammatory markers [119].

Similar results were obtained using cardiosphere-derived cells (CDCs) that are capable of secreting exosomes [120,121] enriched with specific small RNAs [122,123] and proteins [124] in response to cardiac injury. When delivered acutely post MI, these exosomes can polarize macrophages to a cardioprotective state, suppressing the expression of pro-inflammatory cytokines and promoting efferocytosis. In this context, at least under in vitro conditions, miR-181b seems to be the key mediator of this process [125].

Another promising therapeutic strategy may involve the use of human embryonic stem cell-derived cardiovascular progenitor cells (hESC-CVPCs). Recent data show that their use can induce a reparative phenotype in cardiac macrophages in infarcted hearts through a pathway involving signal transducer and activator of transcription 6 (STAT6). Injection of hESC-CVPCs into acutely infarcted myocardium significantly improves cardiac function and scar formation, reducing inflammatory response and cardiomyocyte apoptosis [114].

As already mentioned, the challenge when using stem cells for therapeutic purposes is mainly related to their limited survival after injection due to failed engraftment, necrosis, and apoptosis [94]. However, even these ungrafted cells may have a role in cardiac regeneration. It has been postulated that the apoptotic transplanted cells can inhibit macrophages and dendritic cells and stimulate regulatory T cells, resulting in the downregulation of both innate and adaptive immunity. The result is reduced fibrosis and an overall improved cardiac outcome [126].

Recent studies have also shown positive effects on cardiac function due to drugs commonly used in clinical practice that are mediated by the modulation of macrophage pro-inflammatory activity. Among these drugs, statins, inhibitors of the liver enzyme β-hydroxy β-methylglutaryl-coenzyme A (HMG-CoA) reductase commonly used in the treatment of hypercholesterolemia, have been found. This pharmacological class has also demonstrated anti-thrombotic and anti-inflammatory properties [127] and can stimulate new myocyte formation after MI [128]. Pravastatin inhibits IFN-γ-induced macrophage activation, while simvastatin interrupts MHC class II interactions between macrophages and the adaptive immune complex [129,130]. In addition, angiotension-converting enzyme inhibitors (ACEi) such as Enalapril have been shown to reduce angiotensin II (ATII)-stimulated monocyte recruitment from splenic reservoirs in a murine model, resulting in a 14% improvement in the ejection fraction following MI [131].

## 9. Conclusions

The available evidence suggests that specific immunity may significantly contribute to the different endogenous regenerative responses to injury in regenerative tissues, as opposed to the non-regenerative response of the adult mammalian heart. Nowadays, a great number of different immune cell types have been investigated in cardiac repair and myocardial remodeling after damage, and the main features of their phenotypes are summarized in Table 1. It remains to be established whether cardiac macrophages (recruited vs. resident) overall modulate cardiac pathologic remodeling after injury by preventing an effective regenerative response. Furthermore, it is unknown whether altering the type of cardiac macrophage response (switching from CCR2^+^ to CCR2^−^ by eliminating CCR2^+^ macrophages) promotes effective myocardial regeneration after injury. These missing data could form the conceptual basis for therapeutic strategies that enhance cardiac tissue-resident CCR2^−^, inhibit CCR2^+^ macrophages, or achieve both; these strategies may also have the potential to achieve anatomical and functional myocardial regeneration after injury. Additionally, investing in this research topic will also help to clarify whether cardiac macrophage-dependent immunity activates the formation of new cardiomyocytes through CSC myogenic differentiation or through the unexpected duplication of pre-existing cardiomyocytes.

## Figures and Tables

**Figure 1 ijms-24-10747-f001:**
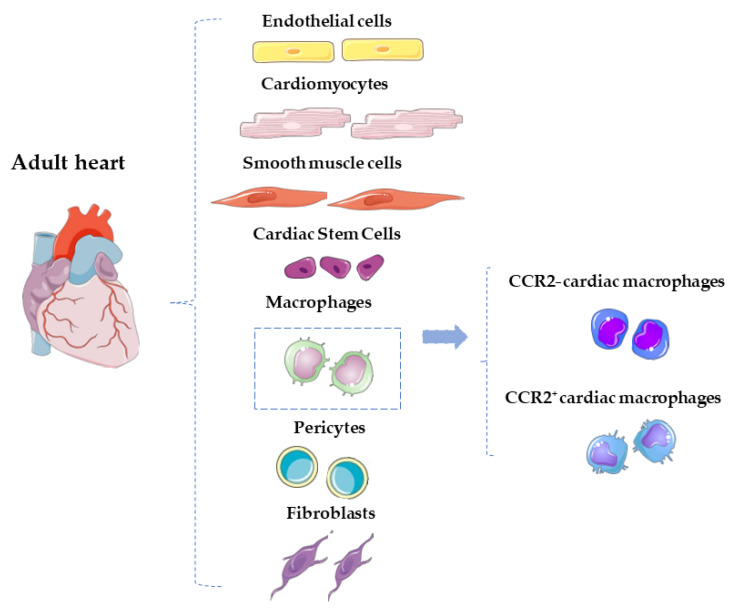
Schematic representation of the main cardiac cellular component of the adult heart. Among the immune cells, macrophages can be distinguished in CCR2^+^ and CCR2^–^ cardiac macrophages.

**Figure 2 ijms-24-10747-f002:**
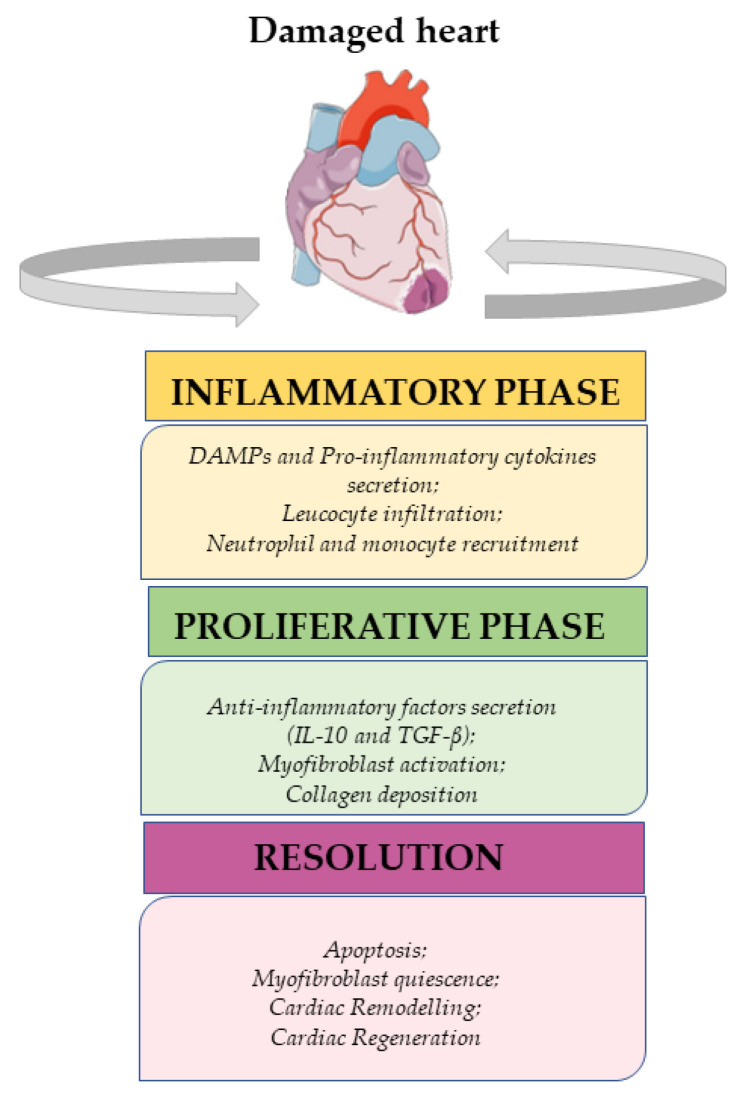
From inflammation to resolution signaling after cardiac damage.

**Figure 3 ijms-24-10747-f003:**
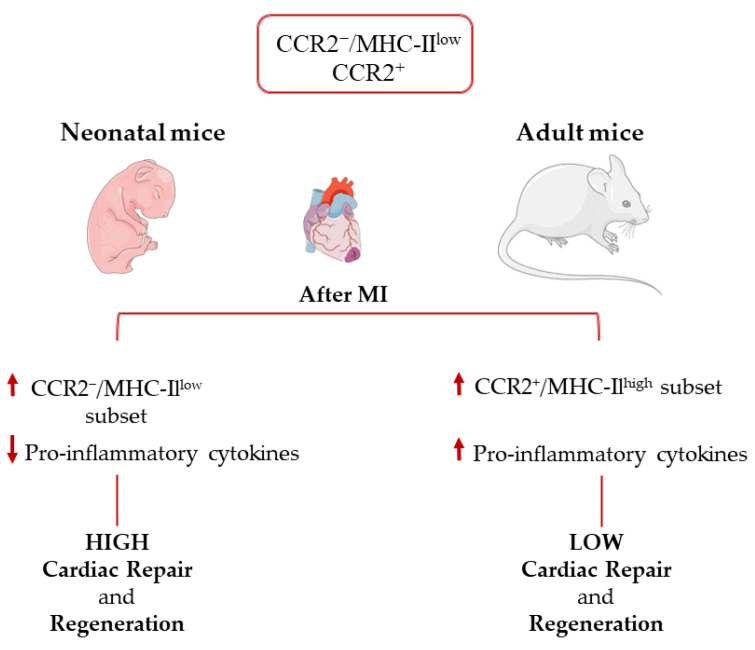
Immune response in the neonatal and adult heart after myocardial infarction.

**Table 1 ijms-24-10747-t001:** Immune cell systems involved in cardiac repair.

Cell Types	Phenotype	Activity	Active Molecules
N1-Neutrophilis	CD11b CD16 CD15 CD87	Degranulation Phagocytosis Apoptosis	MPO; ROS
N2-Neutrophilis	CD11b CD206	Macrophage polarization Angiogenesis Apoptosis	MMP-9; MMP-12, vEGF
Inflammatory Eosinophils	CD62L− CD49d CD101^high^	Degranulation Oxidative stress	Il-5; ROS; EPO
Regulatory Eosinophils	CD62L+ CD101^low^ Sigle-8+	Macrophage polarization and angiogenesis	Il-4; vEGF; FGF; TGF-β
Monocytes	Cd14^+^CD16^−^/Ly-6C^low^ CXC3R1	Secretion of anti-inflammatory and angiogenic cytokines	IL-1β; IL-6; TNFa; NO; TGF-β; vEGF
M1 Macrophages	CCR2 CD68 MHC II CD86	Phagocytosis Secretion of inflammatory cytokines	IL-12, IL-23; ROS, NO
M2 Macrophages	CD206 IL-1Ra TGFβ vEGF	Secretion of anti-inflammatory and angiogenic cytokines	IL-10; IL.1Ra; vEGF; TGF-β
Mast Cells	CD64 CD117	Secretion of inflammatory cytokines Degranulation	Histamine; INF-α; IL-6, IFN-γ
Dendritic Cells	MHC II CD80/CD86	Antigen presentation	IL-23 IL-10
B1- and B2-Lymphocytes	CD19	Secretion of anti-inflammatory cytokines	IgM, IgG
Regulatory B-Lymphocytes	Tim-1	Chemokine production	IL-10
NK-Cells	CD69	Secretion of inflammatory cytokines	CCL7 IFN-γ
T-Lymphocytes	CD4^+^ T-helper CD8^+^ T-cytotoxic CTLA-4 Tregs	Autoreactivity Secretion of inflammatory cytokines	IFN-γ IL-17 TGF-β

## Data Availability

All data are available within this article.

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
