# Peer review of "Polarizing Macrophage Functional Phenotype to Foster Cardiac Regeneration"

_ijms, 2023, doi:10.3390/ijms241310747_

Round 1

Reviewer 1 Report

Molinario et al. reviewed a less touched research area on the roles of macrophages and other immune cells in cardiac regeneration in the context of stem cell- based therapies for cardiac injuries. Although there are some merits in this review, such as novelty, the manuscript lacks focal points. The critical comments are listed below:

1.       The title is not appropriate. The review is mostly about macrophages. There are many other immune cells in innate immune response, such as natural killer cells, neutrophils, mast cells, etc. Neutrophils and granulocytes were briefly mentioned. It is not adequate to use “innate immune response” in the title.

2.       Fig.2 needs a citation.

3.       In Fig. 3, it is interesting that the immune responses after MI are different in neonatal and adult mice. More discussion should be included about any hormonal and epigenetic difference, or even immunosenescence in neonatal and adult mice.

4.       In line 244-246, more details should be added about macrophage subsets in human heart.

5.       A few abbreviations were not properly introduced. For example: MI, LV.

6.       In line 467: “as already mentioned…”, please cite here to help remind the readers where it was mentioned.

7.       There are several “CCR”, probably should be “CCR2”.

8.       In line 422 and 423, “Similarly, also macrophages implicated that….”. Please rewrite this sentence because it is confusing in its current format.

9.       In line 395, “often used interchangeably…” “with aging” should be included.

10.   In line 409, “trough” should be “through”.

11.   In line 127, “as neutrophils” should be “like neutrophils”

12.   In line 104, “facilitate” should be “facilitating”.

The quality of English language is moderate. Some editing is needed.

Author Response

Reviewer #1

Molinaro et al. reviewed a less touched research area on the roles of macrophages and other immune cells in cardiac regeneration in the context of stem cell- based therapies for cardiac injuries. Although there are some merits in this review, such as novelty, the manuscript lacks focal points. The critical comments are listed below:

We thank reviewer #1 for her/his complimentary and constructive comments to our review article. We believe that we have satisfactorily addressed all her/his critical comments as detailed below while also carefully revised the manuscript for English language.

  1. The title is not appropriate. The review is mostly about macrophages. There are many other immune cells in innate immune response, such as natural killer cells, neutrophils, mast cells, etc. Neutrophils and granulocytes were briefly mentioned. It is not adequate to use “innate immune response” in the title.

We understand the point raised by reviewer #1. Therefore, we have changed the titled to reflect her/his advice as: “Polarizing Macrophage Functional Phenotypes to Foster Cardiac Regeneration

  1. Fig.2 needs a citation.

Thanks. Done.

  1. In Fig. 3, it is interesting that the immune responses after MI are different in neonatal and adult mice. More discussion should be included about any hormonal and epigenetic difference, or even immunosenescence in neonatal and adult mice.

We thank reviewer #1 for this relevant point. Indeed, assessing changes in epigenetic regulation through cardiomyocyte development has been of interest because of the drastic change in the cardiomyocyte proliferation ability after the first few days of life (doi: 10.1016/j.stem.2019.12.004). Changes in epigenetics have proven to vary from neonatal proliferative-competent to adult terminally-differentiated cardiomyocytes (doi: 10.1016/j.stem.2019.12.004). Furthermore, loss of heart regenerative capacity in adult versus neonatal mammals is triggered by increasing thyroid hormones and may be a trade-off for the acquisition of endothermy(doi:10.1126/science.aar2038). Finally, with aging, myocardial T cells undergo clonal expansion and exhibit an up-regulated pro-inflammatory transcription signature, marked by an increased interferon-γ (IFN-γ) production, (doi:10.1093/cvr/cvad068). Physiological T-cell development or adoptive transfer of adult IFN-γ-producing T-cells into neonatal infarcted mice shifted them an adult-like healing phenotype with monocyte-derived macrophage recruitment, contributing impaired cardiac regeneration and promoting irreversible structural and functional cardiac damage (doi:10.1093/eurheartj/ehac153). These findings suggest a trade-off between myocardial regenerative potential and the development of T-cell competence as well as postulate that immunosenescence may account for the deficit of regenerative capability with age. We have inserted the discussion of this point in the revised version of our manuscript.

  1. In line 244-246, more details should be added about macrophage subsets in human heart.

Thanks. More details have been accordingly added.

  1. A few abbreviations were not properly introduced. For example: MI, LV.

Thanks. Done.

  1. In line 467: “as already mentioned…”, please cite here to help remind the readers where it was mentioned.

Citation inserted. Thanks.

  1. There are several “CCR”, probably should be “CCR2”.

Corrected. Thanks.

  1. In line 422 and 423, “Similarly, also macrophages implicated that….”. Please rewrite this sentence because it is confusing in its current format.

Corrected. Thanks.

  1. In line 395, “often used interchangeably…” “with aging” should be included.

Inserted. Thanks.

  1. In line 409, “trough” should be “through”.

Corrected. Thanks.

  1. In line 127, “as neutrophils” should be “like neutrophils”

Corrected. Thanks.

  1. In line 104, “facilitate” should be “facilitating”.

Corrected. Thanks.

Reviewer 2 Report

Molinaro and colleagues revised the current literature on the connection between the immune and cardiovascular systems, with a specific focus on the role of macrophages in cardiac injury and repair. The authors highlighted the significance of the immune system in response to cardiac damage, and in the perspective of tissue repair and regeneration. They discuss the different subtypes of macrophages and their influences on cardiac remodeling and repair. Additionally, they emphasize the importance of understanding the innate and adaptive immune systems' differential impact on the response to cardiac injury. The authors also address the unresolved questions surrounding the complex phenotype of cardiac macrophages and their role in stem cell-based therapies for myocardial repair. Finally, their comprehensive overview provides insights into potential novel therapeutic strategies for cardiac regeneration through the modulation of the immune response.

The review is very well written and highly informative for the audience in the field, I only have a minor suggestion. Readers would likely benefit from another table summarizing the different immune cell types involved in cardiac repair.

My compliments to the authors for the excellent work done.

A final minor revision of the english language may be useful.

Author Response

Reviewer #2

Molinaro and colleagues revised the current literature on the connection between the immune and cardiovascular systems, with a specific focus on the role of macrophages in cardiac injury and repair. The authors highlighted the significance of the immune system in response to cardiac damage, and in the perspective of tissue repair and regeneration. They discuss the different subtypes of macrophages and their influences on cardiac remodeling and repair. Additionally, they emphasize the importance of understanding the innate and adaptive immune systems' differential impact on the response to cardiac injury. The authors also address the unresolved questions surrounding the complex phenotype of cardiac macrophages and their role in stem cell-based therapies for myocardial repair. Finally, their comprehensive overview provides insights into potential novel therapeutic strategies for cardiac regeneration through the modulation of the immune response.

The review is very well written and highly informative for the audience in the field, I only have a minor suggestion. Readers would likely benefit from another table summarizing the different immune cell types involved in cardiac repair.

My compliments to the authors for the excellent work done.

We are sincerely grateful to reviewer #2 for her/his complimentary comments. As for her/his suggestion we have now included the Table 1 summarizing the different immune cell types involved in cardiac repair and regeneration.

Round 2

Reviewer 1 Report

The authors have satisfactorily addressed the reviewer's concerns. 

Minor editing would improve the quality of the manuscript.